# Comparative Evaluation of the Performance of Sterile Filters for Bioburden Protection and Final Fill in Biopharmaceutical Processes

**DOI:** 10.3390/membranes12050524

**Published:** 2022-05-16

**Authors:** Jimin Na, Dongwoo Suh, Young Hoon Cho, Youngbin Baek

**Affiliations:** 1Department of Biological Engineering, Inha University, 100 Inha-ro, Michuhol-gu, Incheon 22212, Korea; jmna0227@gmail.com; 2School of Chemical and Biological Engineering, College of Engineering, Institute of Chemical Process (ICP), Seoul National University (SNU), 1 Gwanak-ro, Gwanak-gu, Seoul 08826, Korea; dwsuh1@snu.ac.kr; 3Green Carbon Research Center, Korea Research Institute of Chemical Technology (KRICT), 141 Gajeong-ro, Yuseong-gu, Daejeon 34114, Korea; 4Department of Advanced Materials and Chemical Engineering, University of Science & Technology (UST), Yuseong-gu, Daejeon 34113, Korea

**Keywords:** sterile filter, bacterial challenge test, surface characteristics, filtrate flux, protein adsorption

## Abstract

Sterile filtration processes are widely used in the production of biotherapeutics for microorganism removal and product sterility. Sterile filtration processes can be applied to buffer preparation and cell culture media preparation in biotherapeutics processes, and to final sterilization or final filling in downstream processes. Owing to their broad range of applications in bioprocessing, various 0.2/0.22 μm sterile filters with different polymer materials (i.e., hydrophilic PVDF and PES) and nominal pore sizes are commercially available. The objective of this study was to evaluate two different commercial sterile filters in terms of filtration performance in various sterile filtration processes of biopharmaceutical production. The results demonstrate the importance of choosing the appropriate filter considering the process type and target removal/transport product to ensure efficient sterile filtration in the production of biotherapeutics.

## 1. Introduction

Sterile filtration processes are widely used in the production of biotherapeutics such as monoclonal antibodies (mAbs) or recombinant DNA-derived proteins [1]. In biotherapeutics production, sterile filtration is used in buffer and cell culture media preparations for removing undesired microorganisms [2]. In downstream processes, sterile filtration in the final step, called final filling, is performed to assure product sterility. Final filling is preferred over thermal sterilization as it ensures product stability [3]. Depending on the application, sterile filters with a nominal pore size of 0.1/0.2 μm are used in sterile filtration. For instance, 0.1 μm filters are used for prefiltration, prior to virus filtration for protein aggregate removal [4] or mycoplasma removal [5], whereas 0.2 μm filters are widely applied to remove bioburden and undesired particulates of more than 0.2 μm in size. In the case of 0.2/0.22 μm filters, the filters must demonstrate high retention of *Brevundimonas diminuta (B. diminuta),* known as the bacterial challenge test (BCT) of sterile filtration. The size of *B. diminuta* is approximately 0.3–0.4 μm [6]. In addition to the bacterial challenge, sterile filters must pass various integrity tests to prove their validity in various bioprocesses [7]. 

The American Society for Testing Materials (ASTM) has developed a standard testing method for the bacterial challenge of membranes to be used in liquid filtration [6]. According to the test, the bacterial challenge with *B. diminuta* using 0.2 μm filters must exceed 10^7^ colony forming units (CFU) per square centimeter of the effective filter area. Retention is determined by the difference in bacterial concentration between the feed and permeate solutions. To clear the BCT validation, the model bacteria must be in a non-aggregated and monodispersed state [8]. For 0.1 μm filters, the model microorganism must be changed for accurate retention validation [9].

Commercially available sterile filters are composed of polymers such as polyvinylidene fluoride (PVDF) and polyethersulfone (PES) [1]. In sterile filtration, 100% protein transmission is achieved, whereas protein aggregate deposition or pore blocking causes severe protein fouling, resulting in permeate flux decline and final product loss [10,11,12]. To mitigate this fouling effect during sterile filtration, the filter surface is modified using a hydrophilic coating to reduce protein adsorption [1] or by adding various excipients to reduce protein aggregation [13,14]. In addition to filter fouling by protein, process disruption during media filtration could also occur through undesired intermediates or nutrient aggregate production during the filtration process.

Although various sterile filters are available for biotherapeutic production processes, limited studies have investigated the filtration performance based on different filter materials. Allmendinger et al. [15] showed that PES filters show a higher fouling propensity than PVDF filters due to smaller pore size. However, their study was focused on the effect of protein types and excipient addition on protein fouling during sterile filtration, and not on the filter material. Taylor et al. [16] examined the nanoparticle retention/transmission behavior of various commercial sterile filters. They showed that a narrower pore size distribution (PSD) resulted in the lowest undesired 300 nm particle transmission using 0.2 μm grade sterile filters. The objective of this study was the comprehensive comparison of two different commercial sterile filters regarding bacterial challenge performance, protein transmission, and nutrient medium filtration. In addition to performance, surface characteristics were compared using scanning electron microscopy (SEM), contact angle analysis, X-ray photoelectron spectroscopy (XPS), and attenuated total reflection-Fourier transform infrared (ATR-FTIR) spectroscopy. 

## 2. Materials and Methods

### 2.1. Materials

Two types of 47 mm flat-sheet 0.22 μm sterile filters from the same manufacturer were used in all experiments. *B. diminuta* (American type culture collection, ATCC–19146) was used as the model microorganism for BCT. The model bacterial strain was kindly provided by the Korean agricultural culture collection (KACC). *B. diminuta* was inoculated in 30 mL of nutrient broth (Difco, Detroit, MI, USA) for cell growth up to 10^8^ CFU/mL at 26 °C for 48–72 h. The grown bacterial solution was washed with 10 mM phosphate buffer saline (PBS) solution by cell precipitation using centrifugation at 3000 rpm (i.e., 1000× *g*) for 15 min. The washing process was repeated three times prior to BCT. The final bacterial stock was diluted in model buffer solution to maintain 10^6^–10^7^ CFU/mL. 

In all filtration experiments, 10 mM of Bioperformance-certified PBS (Sigma Aldrich, St. Louis, MO, USA) was used as the model buffer solution. The buffer solution was prepared by dissolving one pack of PBS powder in 1 L of deionized (DI) water and was sterilized by 121 °C autoclaving for 15 min. Luria–Bertani (LB) broth (Difco, USA) was chosen as the model nutrient medium for nutrient medium filtration experiments. The broth powder was reconstituted in DI water to the desired concentration as indicated by the manufacturer. Prior to the filtration experiment, the broth was sterilized by autoclaving for 15 min at 121 °C. Bovine serum albumin (BSA) (Sigma Aldrich, USA) was used as the model protein. BSA powder was dissolved in the model buffer solution to the desired concentration from 1–5 g/L. 

### 2.2. Filter Characteristics

Scanning Electron Microscopy (SEM), X-ray photoelectron spectroscopy (XPS), attenuated total reflection-Fourier-transform infrared (ATR-FTIR) spectroscopy, contact angle analysis using a sessile drop method, and gas–liquid porometry (GLP) were used to analyze the physicochemical properties of the sterile filter surface. Scanning electron microscopy (SEM, SNE-4500 M, SEC, Seoul, Korea) was used to investigate the surface and cross-sectional morphologies of the model sterile filters. An applied voltage of 15 kV and 1000× magnification was adjusted to obtain clear morphologies. For cross-section images, the samples were cut in liquid nitrogen using a sharp knife. 

Surface characteristics were further investigated using Fourier-transform infrared spectroscopy (ATR-FTIR, Alpha-P FTIR spectrometer, Bruker, Billerica, MA, USA) and X-ray photoelectron spectroscopy (XPS, Axis Nova, Shimadzu, Japan) for functional group and surface chemical composition analysis. The average pore size and PSD of sterile filters were analyzed using a capillary flow porometer (Porolux 1000, Porometer NV, Nazareth, Belgium) with Porefil (15 mN/m, Alfa Wessemann Inc., West Caldwell, NJ, USA). To analyze the surface hydrophilicity of model filters, the sessile drop method was used to measure the water contact angle on the filter surface. The contact angle was measured after 5 s and 60 s of the water drop.

### 2.3. Bacterial Challenge Test (BCT)

The bacterial challenge test was conducted based on the ASTM F838-20 guidelines [6], and its experimental procedure is described in Appendix A. Prior to BCT with *B. diminuta*, the negative control test was performed to investigate the overall sterility of the filter. Briefly, 300 mL of pure PBS solution was filtered under 100 kPa. After buffer filtration, the model filters were soaked in nutrient broth and incubated at 30 *±* 2 °C for 7 days to observe bacterial growth after pure solution filtration. After the absence of bacterial growth was observed, the filters were considered sterile. Once the system passed the negative control test, BCT was performed with 300 mL of approximately 10^6^ CFU/mL of *B. diminuta* in 10 mM PBS solution. It is necessary to achieve a bacterial challenge greater than 10^7^ CFU/cm^2^ for use in sterile filtration. The effective filter area for both filters was 14.6 cm^2^. To evaluate BCT, permeate samples were collected and the bacterial concentration was measured. Bacterial retention was calculated using the following Equations (1) and (2).
(1)$$Retentio\;rate\;\left(R\right)=\log_{10}C_{F}-\log_{10}C_{P}$$
where *C_F_* is the feed concentration and *C_P_* is the permeate concentration of bacteria.
(2)$$Retention\;per\;filter\;unit\;area=\frac{\left(R\times V_{p}\right)}{A}$$
where *R* is the retention rate from Equation (1), *V_p_* is the total permeate volume, and *A* is the effective filter area used in the filtration process.

### 2.4. Sterile Filtration Process and Protein Concentration Measurement

Before sterile filtration, all equipment was sterilized by autoclaving and UV irradiation. Model filters were installed in a stainless-steel filter housing with an effective filter area of 1.13 cm^2^. Pressure was applied by N_2_ gas purging. The DI water flux was measured under a pressure range of 20–200 kPa to confirm a linear relationship between the permeate flux and applied pressure. For sterile filtration using PBS, LB broth, and the protein solution, an applied pressure of 60 kPa was determined according to the manufacturer’s suggestion regarding the appropriate pressure for sterile filtration. The weight change in the permeate solution was determined to calculate the permeate flux in L/m^2^/h (LMH). To analyze protein transmission, the filtered permeate of the BSA solution was collected.

The protein concentration was determined using the Coomassie protein assay reagent (Thermo Scientific, Waltham, MA, USA), in the range of 0–25 mg/mL. Protein concentrations higher than 25 mg/mL were diluted ten-fold until the final protein concentration was lower than 25 mg/mL. The protein solution was mixed with the reagent in a ratio of 1:1, and the absorbance of the solution was measured at 595 nm. To analyze protein adsorption on the membrane, model filters after BSA filtration were soaked in 30 mL of PBS solution, followed by excess BSA removal from the filter surface. The adsorbed BSA was detached by ultrasonication and vortexing for 5 min each. 

## 3. Results and Discussion

### 3.1. Characteristics of the Sterile Filters

Figure 1 shows the top, bottom, and cross-section SEM images of filters A and B. The cross-sectional images clearly showed a structural difference between A and B. Filter A had a relatively uniform pore distribution. In contrast, filter B had an asymmetric pore structure, with a highly open porous region having large pores on the top and a retentive region on the bottom. The SEM images were consistent with previously reported data of sterile filters, presenting 0.2 μm pore sizes in retentive sites and homogenous or asymmetric morphologies [17]. Along with the structural differences, a thickness difference was also observed wherein filters A and B had thickness values of 107.7 ± 0.7 μm and 170.7 ± 3.0 μm, respectively. The top and bottom views of filters A and B further demonstrated the different pore structures between them. Filter A showed no structural difference between the top and bottom surfaces, whereas filter B had macro pores on the top surface and 0.2 μm pores on the bottom surface.

Table 1 summarizes the XPS atomic compositions of filters A and B. The atomic concentration of filter A comprised 66.2% carbon, 18.4% fluoride, and 14.5% oxygen along with traces of Si, Zn, and Ca. The atomic concentration of filter B had a carbon, oxygen, and sulfur distribution of 74.2, 22.9, and 2.9%, respectively. Not only was the chemical composition examined using ATR-FTIR, but also major functional groups of sterile filters, as shown in Figure 2. CH_2_ peaks at 2980 and 3024 cm^–1^ in filter A were observed. Another peak of A at 1403 cm^–1^ corresponded to CH_2_ and peaks at 1300–1100 cm^−1^ corresponded to CF_2_–CF_2_ stretching [18]. The major peaks for B such as C–H stretch at 3094 and 3065 cm^−1^, C=C stretches at 1572 and 1487 cm^−1^, S=O bands at 1303 and 1146 cm^−1^ can be observed [19]. The FTIR spectra correlated with the XPS composition data. Based on the XPS and FTIR analysis, filter A was a polyvinylidene fluoride (PVDF) filter and filter B was a polyethersulfone (PES) filter. The measured data showed consistency with previously reported data for PVDF and PES surfaces [20,21]. 

Filter A had a mean pore diameter of 0.3 μm with the highest percent flow measured at a pore diameter of 0.275 μm and Filter B showed a mean pore diameter of 0.26 μm with the highest percent flow observed at 0.253 μm. Slightly larger mean pore diameters were observed for both filters even though their normal pore size is known as 0.22 μm. The data obtained from thickness measurement using Image J with the cross-sectional SEM image (Figure 1c,f) indicated that filter B was thicker (170.7 μm) than filter A (107.7 μm). The thickness of filter B was consistent with the manufacturer’s data, which indicated a thickness range of 150–200 μm whereas a slight discrepancy was observed with filter A when compared with the manufacturer’s data (approximately 125 μm) [22,23]. Surface wettability was evaluated by measuring the contact angle of a water droplet using the sessile drop method (data not shown). The initial contact angle for filter A was 78° and the water drop was fully absorbed by the filter surface in 15 s. In contrast, filter B showed immediate disappearance of the water drop, showing an excellent hydrophilic property. These results indicate that filter B has a more hydrophilic surface than filter A. The characteristics of commercial sterile filters used in the study are summarized in Table 2.

### 3.2. Bacterial Challenge Test

Prior to the BCT, the negative control experiment was performed by soaking pure buffer-flushed filters in nutrient broth for 7 days at 30 ± 2 °C. Both filters showed no bacterial growth for 7 days, which was confirmed by no change in the turbidity of the nutrient broth. Further, no visible precipitation was observed after centrifuging the broth. Based on the negative control test, process sterility was confirmed, and the same filters were used for BCT. Table 3 summarized the BCT results for filters A and B. The results of BCT for both filters satisfied the ASTM guidelines (>7 log_10_/cm^2^) with no bacterial passage, showing excellent sterile performance. 

### 3.3. Filtration Performance Comparison

#### 3.3.1. Water Permeability

Figure 3 shows the water flux vs. applied pressure of filters A and B. To measure water flux, the change in the weight of the permeate sample was recorded every minute until 300 mL of permeate was obtained. The water permeability of filters A and B was 6300 LMH/bar and 17,000 LMH/bar, respectively. The faster water permeability of filter B compared with that of filter A could be explained by the pore structure with hydrophilic surface characteristics. The asymmetric filter structure with the extremely open and porous region of filter B enhanced the solution permeability [17]. In addition to the structural differences, the hydrophilic surface of filter B facilitated faster water transport than that of filter A [24]. 

#### 3.3.2. Capability as the Buffer Preparation and Media Preparation

Figure 4 shows the filtrate flux behavior during PBS buffer and LB broth filtration at a constant pressure of 60 kPa. Both filters showed no flux decline during 300 mL of PBS filtration, indicating that there were no undesired particulates in the pure buffer solution. The filtrate flux under 60 kPa showed a similar permeate speed as the water flux. However, in the case of the LB broth flux, filtrate flux showed a notable decline of 51% and 12% from its initial flux in filters A and B, respectively. Note that the initial flux of LB broth filtration was 3900 ± 375 LMH for filter A and 10,300 ± 608 LMH for filter B. Decreased flux during LB broth filtration could be caused by undesired intermediate formation or nutrient impurity blocking the filter pore. The present data indicated that filter A had more fouling propensity with LB broth than filter B; however, owing to the low initial flux of filter A, a longer operating time was required to obtain the same volumetric throughput. An additional graph plotted as LB broth flux vs. time showed a similar flux decline rate for both filters A and B (Appendix A). The impact of nutrient load on process disruption was reported in the case of the presence of biomass in the feed solution [25]. Pure broth solution does not contain any biomass prior to prefiltration in the upstream process. However, the nutrient itself could be easily contaminated and grow undesired biological contaminants during the process; the sterility of the broth must thus be guaranteed by sterile filtration. From the flux data for LB broth, the maximum capacity of each filter for LB broth was calculated by *V_max_* analysis using the following equation [26].
(3)$$\frac{t}{V}=\frac{1}{Q_{0}}+\left(\frac{1}{V_{max}}\right)t$$
where *t* is the time, *V* is the permeated volume at t, *Q_0_* is the initial flow rate, and *V_max_* is the calculated maximum filtrate volume. The slope of t/v vs. t was determined as 1/*V_max_* (Appendix A), and the calculated *V_max_* was 1667 L and 5000 L for filters A and B, respectively. The higher filter capacity was expected owing to the higher filtrate flux of filter B [17,26]. Based on the LB broth data, the fouling propensity seemed higher with filter A, regardless of a lower initial flux compared with that of filter B. 

#### 3.3.3. Capability as the Final Filling Step in Downstream Processes

Figure 5 shows the flux decline percentage of 1, 2, and 5 g/L BSA filtration under an applied pressure of 60 kPa for a filtration time of 10 min. Pure buffer and broth filtration was performed to investigate the capability of sterile filters in upstream processes; this experiment was then conducted to investigate the capability of model filters for the final filling stage in downstream processes. For both filters, the flux decline rate was increased as the protein concentration increased. In the case of 5 g/L filtration, both filters showed a rapid flux decline of more than 60% in 20 min. For the initial 10 min of filtration, the flux decline rate of filter A was 4.9, 6.4, and 28.0% for 1, 2, and 5 g/L BSA, respectively. Under the same filtration times, filter B showed a more severe flux decline of 12.2, 17.1, and 76.6% for 1, 2, and 5 g/L BSA, respectively. For both filters, 5 g/L filtration showed almost complete flux decline (Appendix A). Severe flux decline with increased protein concentration was expected owing to increased BSA adsorption on the filter [27] and increased solution viscosity [10]. The recent technology of downstream processes has enabled concentrations of proteins up to 100 g/L via single-pass tangential flow filtration (SPTFF) [28]; therefore, a significant flux decline caused by a high protein concentration in sterile filtration could be a crucial factor in unexpected product loss and longer operating times. Therefore, it is important to develop a strategy for preventing flux decline during sterile filtration. Adding excipients could reduce potential protein aggregation [29] and prevent undesired flux decline. Lay et al. [30] found that electrically charged MF membranes could mitigate the fouling effect of counter-ion charged substances. For stable sterile filtration, further investigations are needed to understand the flux decline mechanism of high concentration products and the effective prevention methods. 

The more rapid flux decline with filter B than that with filter A could have resulted from a higher permeate drag force [31,32], rather than from chemical interactions or filter morphologies. Table 4 summarized the protein adsorption of filters A and B after BSA filtration. With regard to chemical interaction, more hydrophilic surfaces were expected to show less protein adsorption because of lower hydrophobic interaction between the protein and filter surface; however, filter B showed more protein adsorption than filter A at all protein concentrations, regardless of having a relatively hydrophilic surface. To exclude the effect of hydrodynamic drag force, further experiments were conducted by adjusting the initial flux of both filters by changing the applied pressure at 150 kPa for filter A. The initial flux of both filters was 10,444 ± 173 LMH. By adjusting the permeate drag force, the flux decline difference was 5%, and showed a similar flux decline behavior to that under constant pressure operation (Appendix A). The experimental data confirmed that the effect of flux decline in sterile filtration was dominated by permeate drag force. Therefore, it is necessary to find the appropriate permeation rate for optimizing sterile filtration without process disruptions.

## 4. Conclusions

In this study, commercial sterile filters made of hydrophilic PVDF and PES were compared with respect to performance as well as filtration characteristics. Both sterile filters showed sterility and high retention validity by passing the BCT. The filter characteristics showed differences in terms of morphology, mean pore diameter, and hydrophilicity. Filter B with an asymmetric filter morphology with a smaller mean pore diameter in the retentive region and a relatively hydrophilic surface achieved higher permeability than filter A. It also showed a higher capacity for LB broth filtration. The flux decline phenomenon with LB broth was mainly caused by concentration polarization or undesired particulate formation. However, the rate of flux decline by nutrient media needs further investigation with respect to the adsorption mechanism of nutrients on the filter surface. Protein filtration revealed a higher fouling propensity with filter B than with filter A under constant pressure operation owing to the higher permeation drag force. The dominant effect of permeate drag force was confirmed by adjusting the initial flux of both filters, which resulted in a similar flux decline. In the case of the same operation time, filter B was preferable because of its higher capacity. Filter A was preferable for the final fill of biopharmaceutical products owing to its low protein adsorption characteristics caused by low permeate flux. Further investigation will be needed to find effective factors of sterile filtration for each use. Overall, this study provides important insights into the selection of sterile filters depending on their use in biopharmaceutical manufacturing.

## Figures and Tables

**Figure 1 membranes-12-00524-f001:**
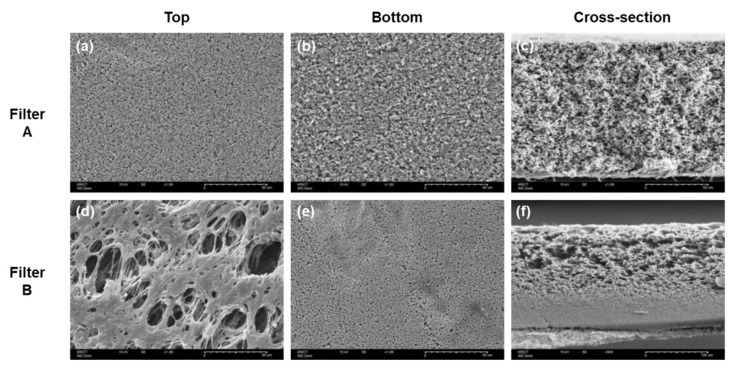
SEM images of filters A (**a**–**c**) and B (**d**–**f**). Top view (**a**,**d**), bottom view (**b**,**e**), and cross-section view (**c**,**f**).

**Figure 2 membranes-12-00524-f002:**
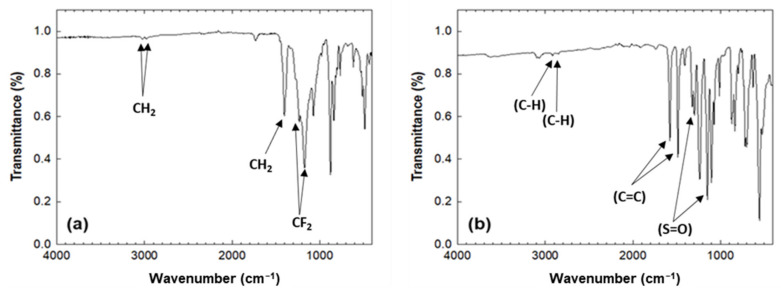
ATR-FTIR spectra for (**a**) filter A and (**b**) filter B.

**Figure 3 membranes-12-00524-f003:**
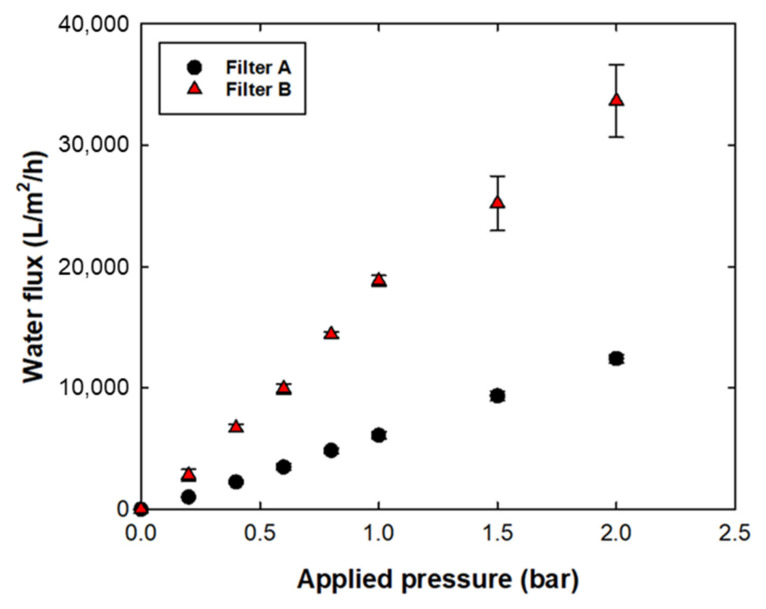
Water permeability of filters A and B.

**Figure 4 membranes-12-00524-f004:**
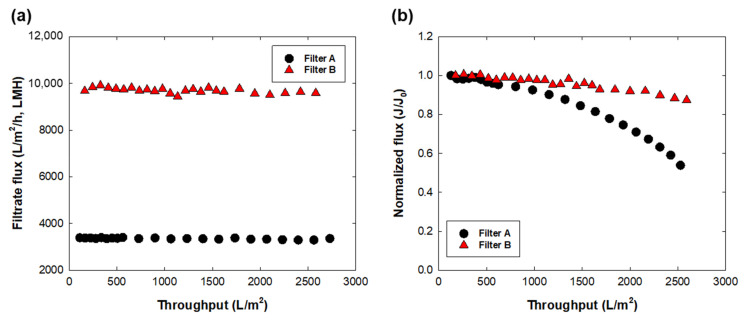
Filtrate flux behavior for (**a**) PBS buffer and (**b**) LB broth with filters A and B. The initial flux (J_0_) was 3400 LMH for filter A and 9700 LMH for filter B.

**Figure 5 membranes-12-00524-f005:**
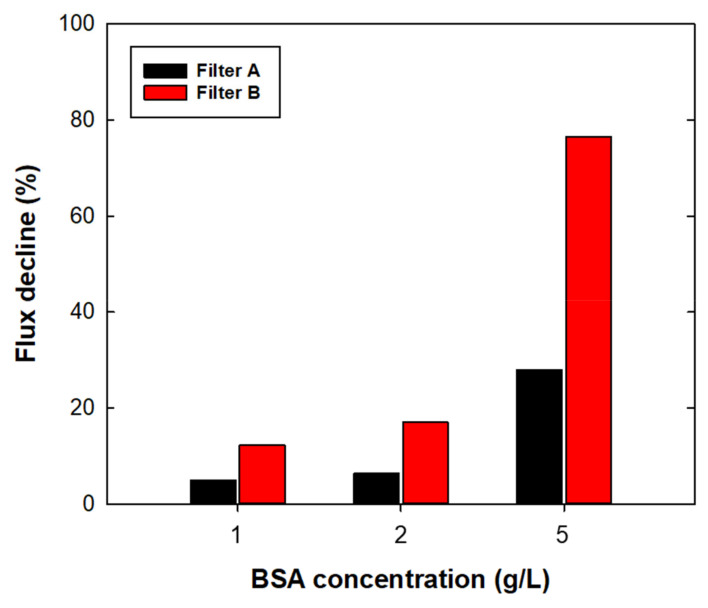
Flux decline (%) as a function of BSA concentration at 1, 2, and 5 g/L using filters A and B for the final fill. Values were obtained from BSA flux data at 10 min of operation time.

**Table 1 membranes-12-00524-t001:** XPS atomic compositions of filters A and B.

	Atomic Concentration (%)						
	C 1s	O 1s	F 1s	S 2p	Si 2p	Zn 2p	Ca 2p
Filter A	66.2	14.5	18.4	-	0.5	0.2	0.2
Filter B	74.2	22.9	-	2.9	-	-	-

**Table 2 membranes-12-00524-t002:** Characteristics of commercial sterile filters.

Model	Material	Measured Pore Size (μm) ^a^	Thickness ^b^ (μm)	Contact Angle (deg)
A	Hydrophilic PVDF	0.3	107.7 ± 0.7	78
B	PES	0.26	170.7 ± 3.0	N.D.

^a^ mean flow pore diameter, measured using gas–liquid porometry (GLP) shown in Appendix A. ^b^ measured from the cross-section SEM image (Figure 1c,f) using the Image J program.

**Table 3 membranes-12-00524-t003:** Results of the bacterial challenge test.

Filter	Feed for Sample (×10^6^ CFU/mL)	Permeate for Sample	Retention Rate (log/cm^2^)
A	4.2	N.D.	>7.8
B	4.0	N.D.	>7.9

**Table 4 membranes-12-00524-t004:** Adsorbed protein concentration onto the filter after the final fill.

Feed BSA Conc. (g/L)	Adsorbed Protein Conc. (µg/cm^2^)	
	A	B
1	N.D.	80.7
2	66.6	132.0
5	152.2	263.2

## Data Availability

Not applicable.

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
