# Peer review of "Comparative Evaluation of the Performance of Sterile Filters for Bioburden Protection and Final Fill in Biopharmaceutical Processes"

_membranes, 2022, doi:10.3390/membranes12050524_

Round 1

Reviewer 1 Report

The manuscript “Comparative Evaluation of the Performance of Sterile Filters for Bioburden Protection and Final Fill in Biopharmaceutical Processes” brings an interesting comparison between PVDF and PES filters applied to upstream processes for sterile filtration of buffer solution and culture medium, and to final filtration of protein solution. It is a valuable contribution to the Bioprocessing field, but the novelty of this report is not clear. Different structures, chemical materials and pore size distribution are documented by the filter manufacturers, which also recommended a particular filter for each application. Do the findings of this manuscript agree or disagree with filter manufacturer recommendations?

The methods for bioburden, flux, protein adsorption, etc., do not appear novel too. If the methods are improved or new, the authors should emphasize it.

Other specific comments are below:

Lines 32-33 The sentence about filtration in upstream process is not precise and needs correction. Sterile filtration and prefiltration are very different. While the primary goal of sterile filtration is to eliminate microorganisms, prefiltration serves to remove cell debris and insoluble particulates not only from nutrient medium, but also from culture broth (if the product is extracellular) or from homogenate after cell lysis.

Lines 239-241 Although the text states that “filter A had more fouling propensity with LB broth than filter B”, the data showed the opposite: the filtrate flux dropped 12% in filter B and only 51% in filter A. Thus, filter B had more fouling propensity. Considering the initial flux of 10,300 LMH, the final flux of filter B was 1,236 LHM (12%). For filter A, the initial flux of 3,900 LMH dropped to 1,989 LHM (51%), still higher than the final flux of filter B. Indeed, the authors concluded the same in lines 255-256: “Based on the LB broth data, the fouling propensity seemed higher with filter A, regardless of a lower initial flux compared with that of filter B”. Therefore, lines 239-241 should be corrected.

Line 123 bacterial growth instead of bacteria growth (L is missing)

Line 204 I did not find the letter “c” in Table 2.

Author Response

The manuscript “Comparative Evaluation of the Performance of Sterile Filters for Bioburden Protection and Final Fill in Biopharmaceutical Processes” brings an interesting comparison between PVDF and PES filters applied to upstream processes for sterile filtration of buffer solution and culture medium, and to final filtration of protein solution. It is a valuable contribution to the Bioprocessing field, but the novelty of this report is not clear. Different structures, chemical materials and pore size distribution are documented by the filter manufacturers, which also recommended a particular filter for each application. Do the findings of this manuscript agree or disagree with filter manufacturer recommendations? The methods for bioburden, flux, protein adsorption, etc., do not appear novel too. If the methods are improved or new, the authors should emphasize it.

We appreciate referee’s valuable comments. This study aimed to evaluate two different commercial sterile filters in terms of filtration performance for bioburden protection and final fill in biopharmaceutical production. We mostly agree with filter manufacturer recommendation which was proved by experimental data in this study. Especially, protein adsorption test along with detailed methods was not provided from manufacturers’. Although previous studies and manufacturers have reported the characteristics of sterile filters, this study presented detailed surface characteristics such as chemical composition, contact angle and filter morphologies with direct comparison in purpose of bioburden protection and final fill. We have revised as recommended by the referee as follows.

Other specific comments are below:

  1. Lines 32-33 The sentence about filtration in upstream process is not precise and needs correction. Sterile filtration and prefiltration are very different. While the primary goal of sterile filtration is to eliminate microorganisms, prefiltration serves to remove cell debris and insoluble particulates not only from nutrient medium, but also from culture broth (if the product is extracellular) or from homogenate after cell lysis.

Thank you for correcting it. We have now revised the sentence to avoid misleading.

“Sterile filtration process can be applied to buffer/media filtration and to final sterilization or fi-nal filling in downstream processes” (line 18-19)

“In biotherapeutics production, sterile filtration is served as buffer and cell culture media preparations for removing undesired microorganism.”  (line 31-33)

Also, the section title “3.3.2. Capability as Prefilters” has been changed as “3.3.2. Capability as the Buffer and Media Preparations” as recommended.

  1. Lines 239-241 Although the text states that “filter A had more fouling propensity with LB broth than filter B”, the data showed the opposite: the filtrate flux dropped 12% in filter B and only 51% in filter A. Thus, filter B had more fouling propensity. Considering the initial flux of 10,300 LMH, the final flux of filter B was 1,236 LHM (12%). For filter A, the initial flux of 3,900 LMH dropped to 1,989 LHM (51%), still higher than the final flux of filter B. Indeed, the authors concluded the same in lines 255-256: “Based on the LB broth data, the fouling propensity seemed higher with filter A, regardless of a lower initial flux compared with that of filter B”. Therefore, lines 239-241 should be corrected.

Filtrate flux of filter B was slightly decreased from 10,300 LMH to ~9,100 LMH, which was 12% decreased flux based on the initial flux. More decreased flux with filter A indicates more fouling propensity. Likewise, similar flux decline was observed when plotting flux vs time. This result is identical because filter A had much lower initial flux.

  1. Line 123 bacterial growth instead of bacteria growth (L is missing)

Thank you for catching this. The relevant sentence was corrected (line 124-125).

  1. Line 204 I did not find the letter “c” in Table 2.

We appreciate for correcting this. The “c” has been removed.

Reviewer 2 Report

An article of performance evaluation and comparison of two different commercial sterile filters was presented. This article is quite interesting and can be potential to provide positive contribution in the field of biopharmaceutical research. However, I recommend the authors to address my comments given below:

  1. The aim of the study should be clearly stated in the abstract and introduction section
  2. The authors mentioned that commercially available sterile filters are composed of polymers such as PVDF, mixed cellulose ester (MCE), and PES. Why is this study only evaluated PVDF and PES?
  3. Line 100. “spectroscopy (XPS)” should be written as “X-ray photoelectron spectroscopy (XPS)” or just “XPS”.
  4. Is there any consideration of choosing B.diminuta in the bacterial challenge test since other study reported broad diversity of viable bacteria in 'sterile' filtered water such Bacteroidetes, Alphaproteobacteria, Betaproteobacteria, Actinobacteria and Spirochaetes? (See https://doi.org/10.1016/j.resmic.2004.05.003)
  5. Author have studied the characteristics of the sterile filters using SEM. In this section, authors just presented the results with no further explanation and discussion regarding to the results. I recommend authors to make it more informative with discussion strengthen with related previous studies cited in the manuscript.
  6. The results of ATR-FTIR spectra in the mode of transmittance were presented. The authors stated that “The chemical composition on the surface was further examined using ATR-FTIR as shown in Figure 2.” But I did not find any further chemical component examination, except the observation of sulfone peaks in both sterile filters. Authors are suggested to provide more comprehensive data discussion and evaluation of dominant peaks found in the ATR-FTIR spectra incorporated with related functional groups (can be presented in table) to enhance the results and discussion.

Author Response

An article of performance evaluation and comparison of two different commercial sterile filters was presented. This article is quite interesting and can be potential to provide positive contribution in the field of biopharmaceutical research. However, I recommend the authors to address my comments given below:

             We appreciate referee’s valuable comments. We have revised as recommended by the referee as follows.

  1. The aim of the study should be clearly stated in the abstract and introduction section

We have now revised to state the objective of this study more clearly.

Abstract (line 21-23): “The objective of this study was to evaluation two different commercial sterile filters in terms of filtration performance in various sterile filtration processes of biopharmaceutical production.”

Introduction (line 70-73): “The objective of this study was comprehensive comparison of two different commercial sterile filters with regard to bacterial challenge performance, protein transmission, and nutrient medium filtration.”

  1. The authors mentioned that commercially available sterile filters are composed of polymers such as PVDF, mixed cellulose ester (MCE), and PES. Why is this study only evaluated PVDF and PES?

Sorry for the misleading. Commercially available sterile filters are composed of polymers such as PVDF, MCE, PES, Nylon and PTFE. In this study, we evaluated PVDF and PES filters which is mostly used as sterile filters in biopharmaceutical processes. We have revised the relevant sentence (line 52-54; 70-73)

  1. Line 100. “spectroscopy (XPS)” should be written as “X-ray photoelectron spectroscopy (XPS)” or just “XPS”.

Thank you for catching this. The terminology has been corrected (line 100).

  1. Is there any consideration of choosing B.diminuta in the bacterial challenge test since other study reported broad diversity of viable bacteria in 'sterile' filtered water such Bacteroidetes, Alphaproteobacteria, Betaproteobacteria, Actinobacteria and Spirochaetes? (See https://doi.org/10.1016/j.resmic.2004.05.003)

We appreciate the referee for suggesting an useful paper regarding bacterial challenge test in sterile filter validation. This study was followed the guideline of bacterial challenge test (BCT) from the American Society for Testing Materials (ASTM) F838-20, which used B. Diminuta as model bacteria for 0.2/0.22 μm grade sterile filters. However, it would be very interesting to further study whether related bacteria exist in biopharmaceutical process.

  1. Author have studied the characteristics of the sterile filters using SEM. In this section, authors just presented the results with no further explanation and discussion regarding to the results. I recommend authors to make it more informative with discussion strengthen with related previous studies cited in the manuscript.

As per referee’s recommendation, we have added more discussion (line 160-164).

  1. The results of ATR-FTIR spectra in the mode of transmittance were presented. The authors stated that “The chemical composition on the surface was further examined using ATR-FTIR as shown in Figure 2.” But I did not find any further chemical component examination, except the observation of sulfone peaks in both sterile filters. Authors are suggested to provide more comprehensive data discussion and evaluation of dominant peaks found in the ATR-FTIR spectra incorporated with related functional groups (can be presented in table) to enhance the results and discussion.

We have now added more comprehensive data with revising Figure 2 to present functional groups for each peak of ATR-FTIR spectra (line 177-183).

Reviewer 3 Report

Dear Authors,

I gave you the highest marks for your paper and your presentation of the data.

Despite it - I require from you the major revision of your paper.

You can ask - why?

In your paper you described two different memberanes - and you did the essential studies which allow to present materials properties and possible applications. You also presented in clearly way your methods of analysis - so each person can perform similar research for each interensting membranes.

However - in you paper you have hidden all data about the manufacturer of these membranes and their models under which they are available on the market. You also have not written why you are or feel compelled to behave in this way.

I know many articles, in which similar studies on other memberans are described - and always the authors presented for readers full data on the manufacturer of membranes and their charanteristics. Thanks to this, recipients can know more about a given product - already tested - and are not forced to repeat the entire cycle of tests.

For example, you can check out one of the recent articles of Prof. Andrew L. Zydney's group

(note to the Editorial Board and the authors - I'm not related with this research group and this research - I was just looking for the first available article from the last year on this subject)

Retention and Fouling during Nanoparticle Filtration:
Implications for Membrane Purification of Biotherapeutics
Liang-Kai Chu 1 , S. Ranil Wickramasinghe 2, Xianghong Qian 3 and Andrew L. Zydney Membranes 2022, 12, 299. https://doi.org/10.3390/membranes12030299

-------

Additional question for the methodology in your paper:

what kind of water you used in your wetting tests? tap water / deionized water / destiled water or pure water (>18.2 MOhm Mili-Q).

Author Response

Dear Authors,

I gave you the highest marks for your paper and your presentation of the data.Despite it - I require from you the major revision of your paper.You can ask - why?

In your paper you described two different memberanes - and you did the essential studies which allow to present materials properties and possible applications. You also presented in clearly way your methods of analysis - so each person can perform similar research for each interensting membranes.

However - in you paper you have hidden all data about the manufacturer of these membranes and their models under which they are available on the market. You also have not written why you are or feel compelled to behave in this way.

I know many articles, in which similar studies on other memberans are described - and always the authors presented for readers full data on the manufacturer of membranes and their charanteristics. Thanks to this, recipients can know more about a given product - already tested - and are not forced to repeat the entire cycle of tests.

For example, you can check out one of the recent articles of Prof. Andrew L. Zydney's group

(note to the Editorial Board and the authors - I'm not related with this research group and this research - I was just looking for the first available article from the last year on this subject)

Retention and Fouling during Nanoparticle Filtration:
Implications for Membrane Purification of Biotherapeutics
Liang-Kai Chu 1 , S. Ranil Wickramasinghe 2, Xianghong Qian 3 and Andrew L. Zydney Membranes 2022, 12, 299. https://doi.org/10.3390/membranes12030299

We appreciate referee’s valuable comments. We used representative symbol instead of specifying the model name because the objective of this study was to evaluate the performance of filters related to their characteristics (not a specific model). In addition, as far as we know, the commercial membranes have been gradually developed and they sometimes kept a model name as same.

-------

Additional question for the methodology in your paper:

what kind of water you used in your wetting tests? tap water / deionized water / destiled water or pure water (>18.2 MOhm Mili-Q).

We used deionized water (Milli-Q® Direct 8 water purification system, Merck Millipore). The relevant sentence has been revised accordingly (line 115-116).

Round 2

Reviewer 2 Report

This manuscript has been revised and showed significant merit compared to the previous version. This version can be accepted only after minor checking on references list. Make sure that the bibliography format is met the requirements to the journal. The URL address of references from internet source should be specific along with the date accessed. 

Author Response

We appreciate referee’s valuable comment on the paper. We have now added accessed date of URL address of references from internet source (line 396-399).

Reviewer 3 Report

Dear Authors

thanks for the manuscript improvements

I now agree with you that the commenrcial name of the filter is not necessary - because in paper we can find the information about the cruccila difference between the filters - the filters materials - PVDF or PES

However, it will be nice to the reader - if you - before the pubilcation - highlight this information in your paper. Reader can find it only in line 177-180 and table 2.

I think it should be incorporate also in Abstract and conclusion 

Not only this - you present the PVDF as hydrophylic surface. And  you said nothing about the PES wettability. Next, you concluce that differences in results came from "differences in terms of morphology, mean pore diameter, and hydrophilicity".

I agree with your conclusion, but I must dicuss with your results and analysis:

PVDF - is one of the most extensively used polymers for superhydrophobic coatings because of its low surface free energy and inertness to various chemicals. ... You also measured the contact angle degree 78 (line 197 and table 2 - what confirm the relatively high hydrophobicity.

In contrary PES is known as  materials with excellent hydrophilic property.

meanwhile, you define PVDF as a hydrophilic material and say nothing about the wettability of PES. I believe that you need to address this problem and make appropriate corrections in your article and comment on this problem, so that readers have no doubts about the compliance (key parameter) of these membranes

I have no other comments - and I agree that the name of the producer company of filters can be not mentioned in paper

Author Response

We appreciate referee’s valuable comment on the paper. As per referee’s comment, we have now added material information in Abstract and Conclusion (line 21; line 312).

PVDF filter used in this study was more hydrophilic than pristine PVDF (usually >120° of contact angle). As mentioned, PES is hydrophilic material so that the result showed immediate disappearance of water drop within 2~3 seconds (it is difficult to determine the contact angle). This results indicate that filter B is more hydrophilic than A. The relevant sentences have been revised accordingly (line 204-206).

Round 3

Reviewer 3 Report

Dear Authors

Thanks for yours paper improving